# PALEO: A PERFORMANCE MODEL FOR DEEP NEURAL NETWORKS

**Hang Qi**
UCLA
hangqi@cs.ucla.edu

**Evan R. Sparks**
UC Berkeley
sparks@cs.berkeley.edu

**Ameet Talwalkar**
UCLA
ameet@cs.ucla.edu

## ABSTRACT

Although various scalable deep learning software packages have been proposed, it remains unclear how to best leverage parallel and distributed computing infrastructure to accelerate their training and deployment. Moreover, the effectiveness of existing parallel and distributed systems varies widely based on the neural network architecture and dataset under consideration. In order to efficiently explore the space of scalable deep learning systems and quickly diagnose their effectiveness for a given problem instance, we introduce an analytical performance model called PALEO. Our key observation is that a neural network architecture carries with it a declarative specification of the computational requirements associated with its training and evaluation. By extracting these requirements from a given architecture and mapping them to a specific point within the design space of software, hardware and communication strategies, PALEO can efficiently and accurately model the expected scalability and performance of a putative deep learning system. We show that PALEO is robust to the choice of network architecture, hardware, software, communication schemes, and parallelization strategies. We further demonstrate its ability to accurately model various recently published scalability results for CNNs such as NiN, Inception and AlexNet.

## 1 INTRODUCTION

Deep learning has been successfully applied in many areas including natural language processing and computer vision. The scale of modern datasets and the millions to billions of parameters in these deep networks pose new challenges when designing computational systems that leverage parallel and distributed computing. Indeed, several important open questions remain:

- *How fast can we train or evaluate a model on a user's given hardware?*

- *For a given architecture, how can a user best leverage parallel and distributed computation?*

- *How can we design a new neural network architecture that can be trained and evaluated efficiently under common hardware setups?*

In response to these fundamental questions, various software packages and systemshave been painstakingly developed, e.g. DistBelief (Dean et al., 2012), TensorFlow (Abadi et al., 2015), MXNet (Chen et al., 2015), SparkNet (Moritz et al., 2015), FireCaffe (Iandola et al., 2016). Moreover, expensive benchmarking efforts, e.g., Chintala et al. (2016), have performed brute-force profiling on some of these deep learning systems on a handful network architectures.

In this work we aim to tackle these questions by taking an analytical approach to model the performance of arbitrary learning systems. Our work hinges on the observation that a neural network architecture is a declarative specification of the forward and backward propagation steps required for training and deploying the network. However, given this specification, there is a rich design space of algorithms, hardware choices, and communications strategies to most efficiently execute these specifications. We build a novel performance model called PALEO[1] that maps this declarative specification to arbitrary points in this design space to estimate the execution time of training and

---

[1]Open-sourced at https://github.com/TalwalkarLab/paleo.

deploying deep neural networks.[2] PALEO applies broadly to a wide variety of neural network architectures and for arbitrary learning systems within this design space, and thus can serve as a valuable tool for practitioners and developers to answer the questions mentioned above.

## 2 BACKGROUND AND RELATED WORK

Training deep neural networks can be very time and resource consuming, and it is not uncommon for the training of a model to take days across tens or hundreds of machines. Several high-level strategies have been proposed to accelerate this process, and these strategies collectively define the design space considered by PALEO.

**Hardware acceleration** approaches are designed to accelerate the computation of the forward and backward passes and often make use of specialized hardware, such as GPUs (Coates et al., 2013), or more recently custom hardware designed specifically for deep learning (Jouppi, 2016). PALEO accepts constants associated with hardware as input (e.g., peak FLOPS, network bandwidth) and automatically adapts to changes in this input.

**Software acceleration** via specialized libraries, e.g., cuda-convnet (Krizhevsky, 2014a) and cuDNN (Chetlur et al., 2014), and highly-optimized algorithms for commonly used primitives, e.g., Chetlur et al. (2014) and Lavin (2016), can also be used to accelerate deep model training. PALEO dynamically picks among the best available implementation for each layer at execution time.

**Parallelization** is a natural approach to consider, and can involve training a neural network with many computational devices (e.g. CPUs, GPUs) on a single machine, or across a network. There are two major parallelization strategies when it comes to training deep neural network models at scale: data parallelism and model parallelism. In classical data parallel systems, each worker stores an identical copy of the model and computes gradients only on a shard of the training examples, and these gradients are aggregated to update the model. In contrast, model parallel systems shard the model itself across the workers, while the training data may be stored on each worker or sharded across the workers. PALEO models both data and model parallel settings.

**Communication schemes** have also been explored to accelerate incremental model updates across distributed workers. Three of the most common schemes are (Iandola et al., 2016; Zhao & Canny, 2013): (i) the OneToAll scheme has a $2KT$ communication time as a master node must communicate with all $K$ workers individually, where $T$ is the time for communicating data through one link in the network; (ii) the Tree AllReduce scheme takes $2\log_2(K)T$ for weights to be aggregated and broadcasted to all workers following a tree topology; and (iii) the Butterfly AllReduce scheme in which all workers receive aggregated weights in $\log_2(K)T$ using a butterfly network. We restrict the focus of PALEO to distributed communication schemes that return equivalent results to serial executions, and we thus do not consider the recently introduced butterfly mixing scheme of Zhao & Canny (2013), or non-deterministic asynchronous parameter servers.

## 3 PALEO

We now present PALEO, a model for the lean consumption of resources during the training of DNNs. PALEO decomposes the total execution time into computation time and communication time; both are estimated for each pass of a neural network's evaluation given user specified choices within the design space of algorithms, hardware, and communications strategies. Figure 1 illustrates the overall idea. The computation time is calculated from factors including the size of the computation inputs imposed by the network architecture, the complexity of the algorithms and operations involved in the network layers, and the performance of the hardware to be used. The communication time is estimated based on the computational dependencies imposed by the network, the communication bandwidth of the hardware, and the assumed parallelization schemes. Once the network architecture and design space choices are fixed, all of the key factors in PALEO can be derived, and we can estimate execution time without actually implementing the entire network and/or an underlying software package.

---

[2]Training a neural network involves both forward and backward propagation, whereas deploying a trained network on a new data point involves only forward propagation. Thus, estimating the execution time of model training encompasses both model training and deployment, and is the focus of this work.

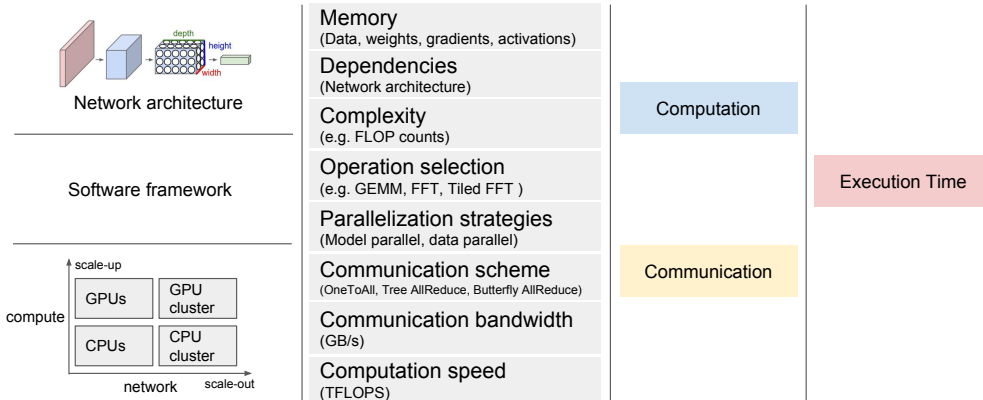

Figure 1: Overview of the PALEO modeling approach. PALEO decomposes execution time into computation time and communication time, which can be derived from various factors implicitly specified by network architectures and hardware configurations.

## 3.1 COMPUTATION MODELING

We first describe the computation model on a single machine. The computation in a neural network can be expressed as a directed graph $\mathcal{N} = \langle \{u^{(i)}\}_{i=1}^{n}, \{(u^{(i)}, u^{(j)})\} \rangle$, where each node $u^{(i)}$ is associated with an operation $f^{(i)}$ on a device $d^{(i)}$; each directed edge $(u^{(i)}, u^{(j)})$ represents the dependency that operation $f^{(j)}$ cannot be executed until $f^{(i)}$ is finished. We use $\text{Pa}(u^{(j)})$ to represent the set of immediate parent nodes of $u^{(j)}$. We model each layer in the neural network as a node, and the connections between layers as edges. In the following text, we omit the superscript index when there is no ambiguity.

### 3.1.1 COMPUTATION TIME FOR A SINGLE LAYER

To model the runtime of a layer $u$, we consider the operation $f$ and decompose the execution time of this operation into three terms (as shown in Figure 2a): the time to fetch the input produced by its parent layers $\mathcal{R}(\text{Pa}(u))$; the time to perform the computation of $f$ on the designated device $d$, i.e., $\mathcal{C}(f, d)$; and the time to write the outputs to the local memory $\mathcal{W}(f, d)$. Assuming a sequential execution, the runtime for a node $u$ can be written as a simple summation:

$$T(u) = \mathcal{R}(\text{Pa}(u)) + \mathcal{C}(f, d) + \mathcal{W}(f, d). \tag{1}$$

Among the three terms, the computation time $\mathcal{C}(f, d)$ is calculated as the FLOP (floating-point operation) counts of the operation divided by the computation speed (FLOPS; floating-point operation per second) of the device: $\mathcal{C}(f, d) = \text{FLOPs}(f)/\text{speed}(d)$. The IO times $\mathcal{R}(\text{Pa}(u))$ and $\mathcal{W}(u)$ are calculated as the size of memory footprints involved in the computation divided by the IO bandwidth of the device. When inputs must be fetched from other devices, e.g. in the case of model parallelism, this IO bandwidth refers to the communication bandwidth between two devices. PALEO treats the speed and bandwidth of devices as parameters given to the model so that users can configure them to reflect user-specific configurations.

Using this per-layer model, we will next describe how to model the computation time of an entire network. We will subsequently we present FLOP counts for layer operations commonly used in modern DNNs in Section 4.

### 3.1.2 COMPUTATION TIME FOR NETWORKS

We first consider simple sequential structures where layers are constructed one after another, as in Figure 2b. The total execution time can be calculated as the sum of execution time of all layers $T(\mathcal{N}) = \sum_{i=1}^{n} T(u^{(i)})$. While this calculation may seem trivial at first glance, it forms the foundation for modeling execution time for more complex architectures.

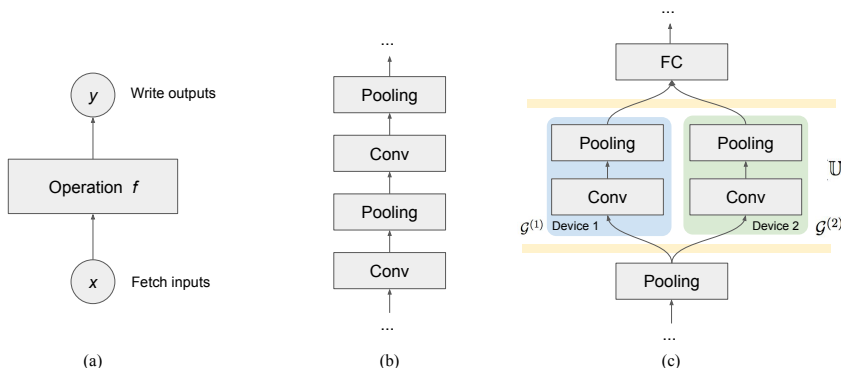

Figure 2: (a) The execution time of a node in the computation graph consists of the time for fetching input, computing results, and writing results to memory. (b) An example of a sequential computation graph segment. (c) An example of a parallel computation graph segment.

Parallel structures are not uncommon in DNNs; for example, the Inception model (Szegedy et al., 2015a) contains layers that can be evaluated simultaneously, and layers on different workers can run in parallel in model parallel setups (Dean et al., 2012). Figure 2c illustrates a parallel structure, where two convolutional layers (each followed by a pooling layer) are scheduled to be executed on two devices.

To model computation time of parallel structures, we identify synchronization barriers before and after every parallel structure and introduce a notation of supernode $\mathbb{U} = \{\mathcal{G}^{(i)}\}_{i=1}^{k}$ as a set of disjoint subgraphs sandwiched by the synchronization barriers in the computation graph. When substituting the subgraphs with the supernode, the network is reduced to a sequential structure described above. For the supernode, the execution time $T(\mathbb{U})$ is within the range $[\max_i T(\mathcal{G}^{(i)}), \sum_i T(\mathcal{G}^{(i)})]$, where the lower bound corresponds to perfect parallelization, the upper bound corresponds to sequential execution. Note that the execution time of a subgraph $T(\mathcal{G}^{(i)})$ can be calculated recursively.

### 3.1.3 COMPUTATION MODELING FOR LAYER OPERATIONS

In modern DNNs, the convolutional layer is one of the most commonly used and computationally intensive type of layer. For this reason, there has been many heavily optimized implementations (Chetlur et al., 2014; Vasilache et al., 2015; Lavin, 2016). Deriving plausible FLOP counts for other types of layers is a straightforward process, and in this section, we consider two leading implementations for convolutional operations: matrix multiplication and Fast Fourier Transform.

Following the notation used by Chetlur et al. (2014), a 2D convolutional layer during forward propagation[3] takes an input feature map $D_{N \times C \times H \times W}$ (which has a batch of $N$ input feature maps with shape $H \times W$ and $C$ channels) and a set of convolutional filters $F_{K \times C \times R \times S}$ ($K$ filters with shape $R \times S$ and $C$ channels). It produces $N \times K$ feature maps each of shape $P \times Q$ which can be calculated from the shapes of inputs and filters together with additional striding and padding parameters. The FLOP counts for the convolution operation can be expressed as $2KCRSNPQ$. A commonly used implementation is to reduce convolution operations to matrix multiplications, which can be efficiently computed with well-optimized SGEMM routines on various platforms. Although these FLOP counts ignore auxiliary operations (e.g. indexing arithmetic in efficient implementations), they nonetheless provide a good estimate of FLOP counts for matrix multiplication implementations.

Another implementation is based on Fast Fourier Transform (Vasilache et al., 2015): both input feature maps and filters are transformed into the frequency domain, then element-wise multiplications are performed followed by an inverse Fourier transform. This implementation introduces computation and memory overhead in discrete Fourier transforms, but reduces the computation complexity to $O(NCKHW + (NC + CK + NK)HW \log(HW))$. Convolutional layers with large filters or a

---

[3]Our arguments generalize to N-dimensional settings, and similar arguments apply for the backward pass.

large problem size can benefit from FFT implementations. When counting FLOPs, it is not possible to get exact counts without knowing the underlying implementation details. In PALEO, we adopt the commonly used FFT complexity $5n \log_2 n$ as the FLOP counts for complex-valued transformations of size $n$ (Cooley & Tukey, 1965). To account for the IO overhead caused by auxiliary memories, PALEO estimates the memory size required for complex-valued matrices in the frequency domain and incorporates it into the data reading and writing terms. For FFT-based implementations with tilings, PALEO estimates the number of tiles from the convolution specifications.

The choice of algorithm – matrix multiplication or FFT – is problem specific, as it depends on the filter size, strides, input size of the convolutional layers, and memory workspace. In order to derive reasonable estimations for user-specific DNNs comparable to real executions, it is important for PALEO to make decisions comparable to real-world systems. Two common approaches are employed in existing DNN software frameworks and libraries to choose between these algorithms: (i) using predefined heuristics based on offline benchmarks; (ii) autotuning to empirically evaluate available algorithms on the given specification. Since autotuning is tied to platform and software implementations, for maximum generality PALEO by default takes the first approach. In particular, PALEO uses heuristics from cuDNN to make algorithm choices while also accounting for user preferences.

## 3.2 COMMUNICATION MODELING

We now describe our modeling for communication among multiple workers. Let $|D|$ be the size of data to be communicated between two workers, and define $B$ as the bandwidth of the communication channel. Then the communication time can simply be written as $T_{\text{comm}} = |D|/B$. By using different bandwidth configurations, PALEO works for both scale-up setups (multiple GPUs on one machine) and scale-out setups (multiple machines in a cluster). Moreover, in data-parallel settings, an AllReduce operation is performed to synchronize model parameters across all workers after every backward pass. PALEO considers three communications schemes: OneToAll, Tree AllReduce, and Butterfly AllReduce. The communication time under these three schemes is described in Section 2.

## 3.3 PLATFORM PERCENT OF PEAK

Thus far, we have assumed that deep learning software platforms make perfect use of their underlying hardware. That is, that the CPUs and GPUs are operating at "peak FLOPS", and that network and IO links are fully saturated. This has allowed our model to be platform independent.

However, this assumption is unreasonable in practice. For instance, achieving peak FLOPS is a difficult proposition, usually requiring customized libraries developed by organizations with intimate knowledge of the underlying hardware, e.g., Intel's MKL (int, 2009), ATLAS (Whaley & Petitet, 2005), and cuDNN. Even these specially tuned libraries may fall short of peak execution by as much as 40% (atl). Further, *any* computation done outside the scope of PALEO (e.g. job scheduling, data copying) will exacerbate the observed inefficiency in practice. Sometimes such inefficiencies are warranted from the perspective of ease of programmability or maintenance of the learning platforms.

Rather than trying to measure and capture every source of inefficiency in every learning framework, we take a small number of representative deep learning workloads which contain convolutions, pooling, dropout, and fully connected layers and run them for a short time on a single GPU. Given observed total throughput and estimated total throughput on this benchmark we fit a scaling constant to estimate a *platform percent of peak* (PPP) parameter which captures the average relative inefficiency of the platform compared to peak FLOPS. Highly specialized frameworks (e.g. cuDNN) will in general have a computational PPP that is close to 100%, while frameworks with higher overheads may have PPP constants closer to 50% or less.

We follow a similar benchmarking procedure to estimate PPP for the communication link for TensorFlow. For the FireCaffe experiments, we estimate the communication PPP based on the empirical results for communication reported in Table 4 of the paper.

## 4 EXPERIMENTS

We now present empirical results which illustrate that PALEO is robust to the choice of network architecture, hardware, communication schemes, and parallelization strategies.

## 4.1 Layer-wise Evaluation

We first compare PALEO-estimated runtimes with actual runtimes measured from Tensor-Flow[4] (Abadi et al., 2015) execution in two popular CNN architectures: the one-tower variant of AlexNet (Krizhevsky, 2014b) and the 16-layer VGG network (Simonyan & Zisserman, 2014). PA-LEO uses cuDNN heuristics to choose algorithms and the auto-tuning mechanism in TensorFlow is disabled. Experiments are run on a NVIDIA TITAN X GPU with a 4 GB workspace limit.

For convolutional and fully connected layers, we evaluate forward computation, backward computation with respect to layer inputs, and backward computation with respect to filters separately (see Figure 4 in the appendix for the plots of layer-by-layer comparison.) Table 1 shows a comparison of full forward pass and backward pass with all layers included. PALEO's per layer estimates are quite close to the actual TensorFlow execution, with only one layer – 'fc6' – consistently being underestimated by PALEO.[5] In spite of this issue with 'fc6', our full pass estimates are remarkably accurate.

Table 1: Full pass time of TensorFlow and PALEO estimation on AlexNet and VGG-16.

|  |  | **Forward pass** (ms) | **Backward pass** (ms) |
|---|---|---|---|
| AlexNet | TensorFlow | 44.00 | 155.10 |
|  | PALEO Estimation | *45.96* | *118.44* |
| VGG-16 | TensorFlow | 400.46 | 1117.48 |
|  | PALEO Estimation | *435.46* | *1077.27* |

## 4.2 Case Studies

We now revisit the questions posed at the beginning of the paper and demonstrate how PALEO can help in answering them. In this subsection we present three case studies. We extract experiment setups including network architectures, hardware specifications, communication schemes, and parallelization strategies from selected publications focusing on scalability of CNNs. We then plug those configurations into PALEO and compare the simulated scalability results with the reported results in the original publications. Table 2 summaries the configurations of PALEO in these experiments.

Table 2: PALEO configurations used in the case studies.

|  | **Case 1** | **Case 2** | **Case 3** |
|---|---|---|---|
| Net | NiN | Inception v3 | AlexNet |
| Device | NVIDIA K20X | NVIDIA K20 | NVIDIA K20 |
| Workers | Up to 128 | Up to 100 | Up to 8 |
| Bandwidth | 70 Gbps | 10 Gbps | 6 GB/s |
| Communication | Tree AllReduce | Parameter Server | Various |
| Parallelization | Data Parallelism | Data Parallelism | Hybrid |
| Platform | Caffe | TensorFlow | cuda-convnet2 |
| **One Step Time**[6] |  |  |  |
| PALEO Estimation | 1918 ms | 4269 ms | 402 ms |
| Reported Time[7] | 2275 ms | – | 418 ms |

---

[4]TensorFlow 0.9 with cuDNN 4 backend.

[5]Examining the TensorFlow execution with the NVIDIA profiler revealed that TensorFlow spent two-thirds of its reported 'fc6' time in transforming data layout between NHWC and NCHW when calling the underlying cuBLAS primitives.

[6]Total time of forward pass, backward pass, and parameter update for one mini-batch on one worker.

[7]Reported times for Cases 1 and 3 are derived approximately from information in the publications. For Case 2 no run time information is provided.

### 4.2.1 CASE 1: NiN WITH FIRECAFFE

FireCaffe (Iandola et al., 2016) adopts the Tree AllReduce communication scheme when training a NiN model (Lin et al., 2013) in data parallel settings with up to 128 servers on the Titan supercomputer. They report a $38\times$ speedup for NiN with batch size 1024 relative to single-GPU performance. Tabel 3 shows the results from PALEO compared with the results reported by FireCaffe.

Table 3: Comparison between PALEO estimation and FireCaffe for training NiN.

| Workers | Batch size | FireCaffe | | PALEO Estimation | |
|---|---|---|---|---|---|
| | | Train Time | Speedup | Train Time | Speedup |
| 1 | 256 | 5.8 days | $1\times$ | *4.9 days* | *1×* |
| 32 | 256 | 11 hours | $13\times$ | *7.6 hours* | *15.5×* |
| 32 | 1024 | 6 hours | $23\times$ | *4.6 hours* | *25.3×* |
| 128 | 1024 | 3.6 hours | $39\times$ | *2.3 hours* | *51.6×* |

### 4.2.2 CASE 2: INCEPTION WITH TENSORFLOW

Murray et al. (2016) reported their results in synchronously training the Inception model (Szegedy et al., 2015b) with TensorFlow and achieved a $56\times$ speedup with 100 workers. They apply a weak scaling strategy with batch size 256 to keep GPUs saturated. Although Murray et al. (2016) leveraged a distributed parameter server rather than one of the three communications schemes considered in PALEO, the communication cost of Butterfly AllReduce can be viewed as a lower bound (Zhao & Canny, 2013). To account for the fact that they train with worker nodes each of which have 8 GPUs, we assumes a linear speedup for GPUs on the same host. Figure 3a shows a comparison between reported speedups and PALEO estimated speedups. For absolute runtime, in one of the experiments, their model completes 20 epochs of training after 100 hours when using 8 Tesla K40's and a batch size 256. PALEO projects a 111 hours runtime under the same setting.

### 4.2.3 CASE 3: ALEXNET WITH HYBRID PARALLELISM

Krizhevsky (2014b) describes a hybrid model and data parallelism approach for training AlexNet using up to 8 GPUs with a weak scaling strategy. In his setup, each of the two CPUs connects to 4 GPUs, the communication bandwidth is penalized by 50% across the two groups as mentioned in the paper. Table 4 shows the comparison between PALEO's projection and the original result, which are quite similar. Moreover, whereas Krizhevsky (2014b) does not quantify the speedup of hybrid parallelism relative to strict data parallelism, PALEO simulates training the entire network with only data parallelism (see last two columns of Table 4) in order to estimate this speedup.

Table 4: Comparison between PALEO estimation and Krizhevsky (2014b) for training AlexNet.

| Workers | One Weird Trick | | PALEO Estimation | | | |
|---|---|---|---|---|---|---|
| | Hybrid parallelism | | Hybrid parallelism | | Data parallelism | |
| | Train Time (h) | Speedup | Train Time (h) | Speedup | Train Time (h) | Speedup |
| 1 | 98.95 | $1\times$ | *96.31* | *1×* | *96.31* | *1×* |
| 2 | 50.24 | $1.95\times$ | *49.57* | *1.94×* | *55.90* | *1.72×* |
| 4 | 26.20 | $3.74\times$ | *25.42* | *3.79×* | *32.82* | *3.03×* |
| 8 | 16.68 | $6.25\times$ | *14.37* | *6.70×* | *23.65* | *5.40×* |

### 4.3 HYPOTHETICAL SETUPS

In this subsection, we use PALEO in two hypothetical setups to analyze the scalability of AlexNet and a GAN model under different communication schemes.

### 4.3.1 ALEXNET IN A CLOUD-BASED SETUP

In this study, we present an analysis of data parallel training of AlexNet. We assume a modern cloud setup with a cluster of servers each equipped with a NVIDIA K80 GPU connected to a 20 Gbps network. In contrast to the Inception model with 23 million parameter, the one-tower variant of AlexNet has 50 million parameters and therefore doubles communication workload when training with data parallelism.

We show strong scaling for all three communication schemes in Figure 3c. Even when assuming a fairly large batch size of 2048 which is beneficial in distributed settings, we see very modest speedups. The OneToAll scheme achieves a max speedup of less than a 2× using 4 workers, while the communication-efficient Butterfly AllReduce scheme achieves a max speedup of roughly 5× when using 32 workers. The weak scaling results, shown in Figure 3b, show drastically improved scaling results, as we observe nearly linear speedups as we increase the number of workers. However, it is important to note that we are increasing the effective batch size as we increase the number of workers, and it is well-known that training with large effective batch-sizes can yield models with substandard accuracy (Breuel, 2015).

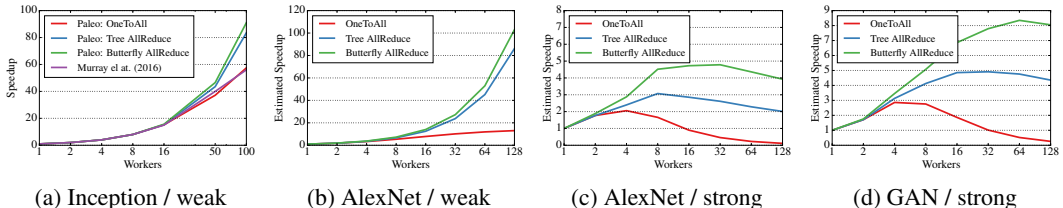

|  (a) Inception / weak | (b) AlexNet / weak | (c) AlexNet / strong | (d) GAN / strong |

Figure 3: Comparison of PALEO projected speedups for various networks under different scaling strategies and communication schemes. (a-b) weak scaling. (c-d) strong scaling.

### 4.3.2 GAN ARCHITECTURE

PALEO can be applied to architectures other than CNNs. We profile a generative adversarial network (GAN) inspired by Radford et al. (2015) for the LSUN dataset with the same hardware assumptions as the previous case study. Table 5 shows that PALEO estimations are close to empirical TensorFlow run time for both the discriminator and generator networks. Figure 3d plots the estimated speedups for training the model with a batch size 256 on up to 128 workers under strong scaling. Without communication-intensive fully-connected layers, while training this GAN architecture is more scalable than AlexNet, PALEO still only predicts an 8× sub-linear speedup with 64 workers.

Table 5: Full pass time of the discriminator and generator in a GAN architecture.

|  |  | **Forward pass** (ms) | **Backward pass** (ms) |
|---|---|---|---|
| Discriminator | TensorFlow | 30.19 | 77.39 |
|  | PALEO Estimation | *27.55* | *79.25* |
| Generator | TensorFlow | 110.11 | 374.18 |
|  | PALEO Estimation | *117.02* | *324.49* |

## 5 CONCLUSION

We introduced PALEO – an analytical performance model for exploring the space of scalable deep learning systems. By extracting computational requirements carried by neural network architectures and mapping them to the design space of software, hardware, and communication strategies, PALEO can effectively and accurately model the expected scalability and performance of a putative deep learning system.

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

# A

We include supplementary figures in appendix due to the space constraint.

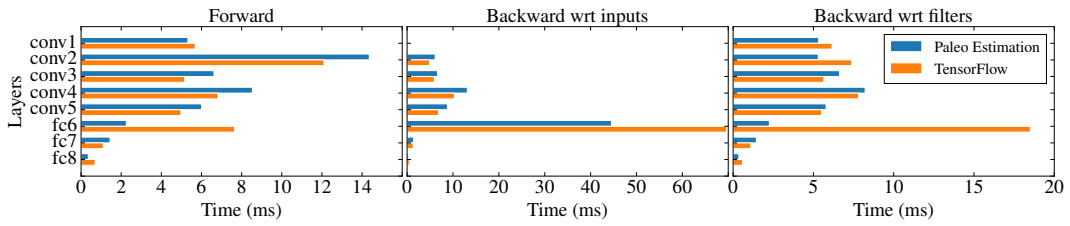

(a) Layer-wise comparison in AlexNet.

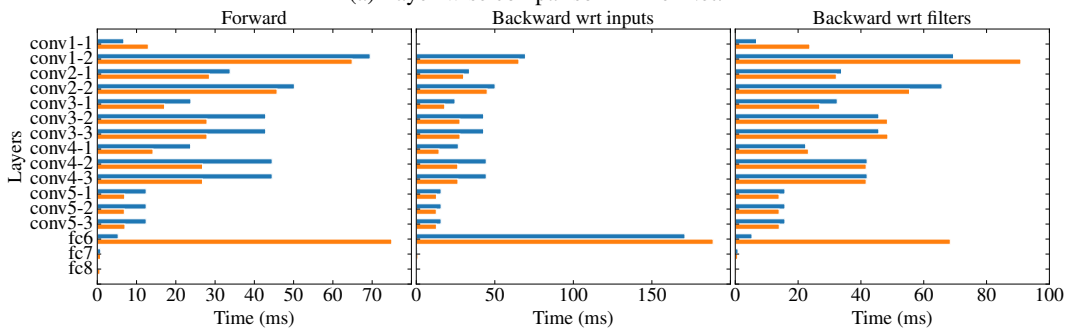

(b) Layer-wise comparison in VGG-16.

Figure 4: Layer-wise comparison between PALEO Estimation and TensorFlow in AlexNet (Krizhevsky, 2014b) and VGG-16 (Simonyan & Zisserman, 2014).

