# Peer review of "Paleo: A Performance Model for Deep Neural Networks"

_ICLR 2017 — accepted_

[Official Review · AnonReviewer3 · rating 6 · confidence 4 · 05 Dec 2016 (modified: 06 Dec 2016)]
**Technically sound. Only useful under the assumption that the code is released.**

This paper is technically sound. It highlights well the strengths and weaknesses of the proposed simplified model.

In terms of impact, its novelty is limited, in the sense that the authors did seemingly the right thing and obtained the expected outcomes. The idea of modeling deep learning computation is not in itself particularly novel. As a companion paper to an open source release of the model, it would meet my bar of acceptance in the same vein as a paper describing a novel dataset, which might not provide groundbreaking insights, yet be generally useful to the community.

In the absence of released code, even if the authors promise to release it soon, I am more ambivalent, since that's where all the value lies. It would also be a different story if the authors had been able to use this framework to make novel architectural decisions that improved training scalability in some way, and incorporated such new insights in the paper.

UPDATED: code is now available. Revised review accordingly.

[Official Review · AnonReviewer2 · rating 6 · confidence 4 · 14 Dec 2016 (modified: 23 Jan 2017)]
**Final review: Sound paper but a very simple model, few experiments at start but more added.**

In PALEO the authors propose a simple model of execution of deep neural networks. It turns out that even this simple model allows to quite accurately predict the computation time for image recognition networks both in single-machine and distributed settings.

The ability to predict network running time is very useful, and the paper shows that even a simple model does it reasonably, which is a strength. But the tests are only performed on a few networks of very similar type (AlexNet, Inception, NiN) and only in a few settings. Much broader experiments, including a variety of models (RNNs, fully connected, adversarial, etc.) in a variety of settings (different batch sizes, layer sizes, node placement on devices, etc.) would probably reveal weaknesses of the proposed very simplified model. This is why this reviewer considers this paper borderline -- it's a first step, but a very basic one and without sufficiently large experimental underpinning.

More experiments were added, so I'm updating my score.

[Official Review · AnonReviewer1 · rating 7 · confidence 4 · 16 Dec 2016]
**No Title**

This paper introduces an analytical performance model to estimate the training and evaluation time of a given network for different software, hardware and communication strategies. 
The paper is very clear.  The authors included many freedoms in the variables while calculating the run-time of a network such as the number of workers, bandwidth, platform, and parallelization strategy. Their results are consistent with the reported results from literature.
Furthermore, their code is open-source and the live demo is looking good. 
The authors mentioned in their comment that they will allow users to upload customized networks and model splits in the coming releases of the interface, then the tool can become very useful.
It would be interesting to see some newer network architectures with skip connections such as ResNet, and DenseNet.

[Author Response · Hang Qi · 12 Jan 2017]
**Rebuttal**

We thank all the reviewers for reading and commenting on the paper! 

AnonReviewer1: “It would be interesting to see some newer network architectures with skip connections such as ResNet, and DenseNet”

Response: 
We released a converter to port Caffe model specs to Paleo, and thus Paleo now supports a wide range of CNNs and residual networks.  Our GitHub repository provides several examples, including ResNet-50 and DenseNet via the Caffe converter. Details can be found in our open source repository (

[Final Decision · Program Chairs · 06 Feb 2017]
**ICLR committee final decision**

The reviewers were consistent in their praise of the paper. They asked for newer architectures, e.g. ResNet, DenseNet. The authors released an update with a Caffe converter which provides access to a wide range of CNNs and residual networks (ResNet-50 and DenseNet examples are provided). This seems like an incredibly useful tool and very glad it is open source. Paper is a clear accept.